# The Association between Caffeine Intake and the Colonic Mucosa-Associated Gut Microbiota in Humans—A Preliminary Investigation

**DOI:** 10.3390/nu15071747

**Published:** 2023-04-03

**Authors:** Annie Dai, Kristi Hoffman, Anthony A. Xu, Shawn Gurwara, Donna L. White, Fasiha Kanwal, Albert Jang, Hashem B. El-Serag, Joseph F. Petrosino, Li Jiao

**Affiliations:** 1Department of Dermatology, Baylor College of Medicine, Houston, TX 77030, USA; 2The Alkek Center for Metagenomics and Microbiome Research, Department of Molecular Virology and Microbiology, Baylor College of Medicine, Houston, TX 77030, USA; 3Department of Medicine, Baylor College of Medicine, Houston, TX 77030, USA; 4Texas Medical Center Digestive Disease Center, Houston, TX 77030, USA; 5Dan L Duncan Comprehensive Cancer Center, Baylor College of Medicine, Houston, TX 77030, USA; 6Center for Innovations in Quality, Effectiveness and Safety, Michael E. DeBakey VA Medical Center, Houston, TX 77021, USA; 7Section of Gastroenterology, Michael E. DeBakey VA Medical Center, Houston, TX 77030, USA; 8Deming Department of Medicine, Tulane University School of Medicine, New Orleans, LA 70112, USA

**Keywords:** phytochemical, coffee, diet, microbiome, *Erysipelatoclostridium*, riboflavin

## Abstract

We examined the association between caffeine and coffee intake and the community composition and structure of colonic microbiota. A total of 34 polyp-free adults donated 97 colonic biopsies. Microbial DNA was sequenced for the 16S rRNA gene V4 region. The amplicon sequence variant was assigned using DADA2 and SILVA. Food consumption was ascertained using a food frequency questionnaire. We compared the relative abundance of taxonomies by low (<82.9 mg) vs. high (≥82.9 mg) caffeine intake and by never or <2 cups vs. 2 cups vs. ≥3 cups coffee intake. False discovery rate-adjusted *p* values (*q* values) <0.05 indicated statistical significance. Multivariable negative binomial regression models were used to estimate the incidence rate ratio and its 95% confidence interval of having a non-zero count of certain bacteria by intake level. Higher caffeine and coffee intake was related to higher alpha diversity (Shannon index *p* < 0.001), higher relative abundance of *Faecalibacterium* and *Alistipes*, and lower relative abundance of *Erysipelatoclostridium* (*q* values < 0.05). After adjustment of vitamin B2 in multivariate analysis, the significant inverse association between *Erysipelatoclostridium* count and caffeine intake remained statistically significant. Our preliminary study could not evaluate other prebiotics in coffee.

## 1. Introduction

Caffeine has been widely consumed in food and drink for centuries. Caffeine is well known for stimulating wakefulness, alertness, and energy. Caffeine has been associated with decreased risk of cardiovascular disease, Parkinson’s disease, and type 2 diabetes mellitus [1,2]. Some gastrointestinal health benefits from caffeine include decreased odds of ulcerative colitis and acute colitis development with in vitro and in vivo studies [3,4], reduced hepatic fibrosis in patients with mild to advanced hepatic fibrosis [5,6], and lower risk of developing colorectal cancer [7].

Caffeine is known to antagonize the adenosine A_1_ and A_2a_ receptors, mobilize intracellular calcium in muscle tissue, and inhibit phosphodiesterases and gamma-aminobutyric acid receptors, leading to increased catecholamine synthesis and turnover in the body and enhanced secretion of dopamine, serotonin, and acetylcholine in the central nervous system [8]. While these effects could explain caffeine’s neuropsychiatric effects, they do not readily explain caffeine’s effects on cancer, cardiovascular disease, liver disease, and diabetes.

Previous studies have suggested that coffee modulates gut microbiota. One study showed that coffee intake decreased the relative abundance of *Prevotella* in fecal samples in a mouse model [9]. In contrast, another study in humans showed that heavy coffee consumers (45–500 mL/daily) had higher fecal *Prevotella* abundance [10]. *Prevotella* has a complex association with inflammation and obesity. Another study showed that 16 adults who consumed three cups of coffee daily for three weeks had an increased abundance of anti-inflammatory *Bifidobacterium* in their feces [11]. However, the association between the intake of phytochemicals in coffee, such as caffeine, and gut microbiota has not been well established in either experimental models or humans.

In this cross-sectional study, we compared the community composition and structure of the colonic adherent microbiota based on caffeine intake using 16S rRNA gene sequencing among 34 individuals with endoscopically normal colons. We hypothesized that individuals with higher caffeine intake could have different microbial community composition and structure compared to those with lower caffeine intake. Because coffee is the major source of caffeine [12], we also evaluated the association between coffee consumption and gut microbiota in the present study. Better knowledge of the association between caffeine and coffee intake and the gut microbiota may help refine dietary guidance.

## 2. Materials and Methods

### 2.1. Study Participants

Participants were recruited at the endoscopy suite of the Michael E. DeBakey VA Medical Center (MEDVAMC) in Houston, Texas, between July 2013 and April 2017. The study cohort, study design, and exclusion criteria were described in detail previously [13]. Individuals were eligible to be enrolled in the study if they were between 50 and 75 years of age. Individuals were ineligible if they had the following: (1) hereditary polyposis syndromes such as familial adenomatous polyposis and hereditary non-polyposis colon cancer; (2) inflammatory bowel disease; (3) invasive cancer except for non-melanoma skin cancer; (4) colorectal polyps in the past three years; (5) end-stage renal disease requiring dialysis; (6) severe mental disabilities; (7) hospitalization within the past year; (8) oral or systemic use of antibiotics in the past three months; (9) hepatitis B or C infections; (10) HIV or methicillin-resistant Staphylococcus aureus-positive infection; or (11) contraindications to obtaining mucosal biopsies.

The study protocol was approved by the Institutional Review Board of Baylor College of Medicine (BCM) and MEDVAMC. All procedures performed in studies involving human participants were in accordance with the ethical standards of the institutional and/or national research committee and with the 1964 Helsinki Declaration and its later amendments or comparable ethical standards. The research coordinator obtained informed consent from participants during their attendance at an educational session 1–2 weeks before the colonoscopy.

### 2.2. Data Collection

After obtaining informed consent, the research coordinator administered a questionnaire collecting information on lifestyle, social history, and medical history. We assessed caffeine intake and the number of cups of coffee consumed in the past 12 months using the validated self-administered Block Food Frequency Questionnaire (FFQ) 2005 [14]. Participants answered questions on daily liquid intake, including “how often did you drink coffee, regular or decaf in the past year” and “how many cups did you drink per day in the past year?”. All categories were combined to arrive at aggregate caffeine intake by Nutritionquest. Caffeine intake (mg/day) was derived from coffee, hot tea, iced tea, and 40 other food items. All diet and nutrient variables were energy-adjusted using the density method. The healthy eating index (HEI) 2015 was used to score dietary quality [15].

### 2.3. Colonoscopy and Biopsy Requirement

Participants were advised to stop taking aspirin, anti-inflammatory drugs, blood thinners, iron, or vitamins with iron seven days before the procedure and to stop diabetic medication one day before the procedure [13]. Participants took 3.78 L of polyethylene glycol (Golytely) the night before the colonoscopy. During the procedure, endoscopists obtained biopsies from each colonic segment (cecum, ascending, transverse, descending, sigmoid colon, or rectum) when possible.

We enrolled 612 eligible participants in the study. Of 174 participants who were confirmed polyp-free, 134 consented to provide colonic mucosal biopsies. Samples from 69 polyp-free participants were sent for microbiota profiling. Among those, 40 returned the Block FFQ. Five study participants who had self-reported energy intake <800 or >5000 kcal per day were excluded from the analysis. As a result, the sequencing data of 99 mucosal samples from 35 participants were available for analysis. A flow chart of participant selection and final sample size is shown in Figure 1.

### 2.4. Microbial DNA Extraction and 16S rRNA Gene Sequencing

The library preparation and sequence analysis were performed at the Alkek Center for Metagenomics and Microbiome Research at BCM. Microbial genomic DNA was extracted from biopsies using the MO BIO PowerLyzer UltraClean Tissue & Cell DNA Isolation Kits (MO BIO Laboratories, Claridad, CA, USA). All DNA samples were stored at −80 °C until further analysis.

The 16S rRNA hypervariable region 4 (V4) was amplified using PCR with the barcoded Illumina adaptor-containing primers 515F and 806R and sequenced on the MiSeq platform (Illumina, San Diego, CA, USA). The 2 × 250 bp paired-end protocol yielded pair-end reads that overlapped almost completely [16,17,18].

### 2.5. Bioinformatics and Taxonomic Assignment

We used the in-house bioinformatics pipeline for data analysis. Reads were merged using USEARCH v7.0.1090 [19]. A quality filter was applied to the merged reads, and those containing more than 0.5% expected errors were discarded. We used the Divisive Amplicon Denoising Algorithm 2 (DADA2) v1.10.1 package in R v3.3.3 to classify the bacteria using the amplicon sequence variants (ASVs). The ASVs were mapped to the SILVA v128 to determine taxonomies [20,21]. A rarefaction curve, using a factor of 4356, was constructed using the sequence data for each sample to ensure that we sampled most of its microbial diversity. Following rarefaction, two mucosal samples with poor sequence reads were lost, including one sample from a participant who contributed one piece of biopsy. Therefore, we included 97 mucosal samples of 34 participants in the final analysis (Figure 1).

### 2.6. Statistical Analysis

Caffeine intake was categorized into higher vs. lower based on the median consumption of the 34 participants, 82.9 mg per day. The standard amount of caffeine in 5 ounces of coffee is 85 mg [22]. Coffee intake was categorized into “<2 cups (16 oz)”, “2 cups”, and “≥3 cups (24 oz)”.

The study participants’ sociodemographic and clinical characteristics and nutrient intake were compared based on caffeine and coffee intake using Fisher’s exact test, Student’s *t* test, or ANOVA. The bacterial alpha diversity (the number of observed OTUs and Shannon Index) and the relative abundance of taxa were compared based on the intake of caffeine and coffee using a Wilcoxon test or Kruskal–Wallis test when appropriate. PERMANOVA with weighted UniFrac dissimilarity and principal-coordinate analysis were carried out for microbial beta diversity analyses [23]. The Monte Carlo permutation test was performed to estimate *p* values. For bacteria with a relative abundance >1% that differed significantly (*q* < 0.05) by both caffeine and coffee intake, we used multivariable negative binomial regression models for panel data to examine the incidence rate ratio (IRR) and its 95% confidence interval (CI) of having non-zero bacterial count in those with higher intake of caffeine or coffee compared with those with a lower intake, adjusting for age (continuous), ethnicity (non-Hispanic white, non-Hispanic African American, and Hispanic), body mass index (BMI) (continuous), smoking (never, former, and current), alcohol use (never, former, and current), HEI score (continuous), and colon segments. Intakes of other nutrients, including total fat, total carbohydrate, total protein, and vitamin B family (vitamin B2, B6, and B12), were also evaluated as potential confounding factors. Vitamin B2 intake was adjusted in the multivariate model in addition to HEI because it differed significantly by caffeine intake and was associated with the relative abundance and count of gut bacteria [24]. Coffee intake was modeled as a continuous variable in the multivariable model. We treated each participant as a panel because some participants contributed multiple biopsies to the analysis.

We used STATA 16.0 (Stata Corp LLC, College Station, TX, USA) and R program for data analysis. All tests were two-sided. A *p* value < 0.05 indicated statistical significance. In the microbiota analysis, all *p* values were adjusted for multiple comparisons using the false discovery rate (FDR) algorithm [25]. FDR *p* values (*q* values) < 0.05 indicated statistical significance.

## 3. Results

The average caffeine intake was 39.2 mg in those who had a “lower intake” and 138.9 mg in those who had a “higher intake”. The differences in the distribution of patients’ characteristics, lifestyle factors, HEI score, and intake of macronutrients by caffeine intake (Table 1) and coffee intake (Appendix A) were largely not statistically significant. Participants who had a higher caffeine intake had a statistically insignificant higher BMI than those who had a lower intake (35.2 vs. 32.6 kg/m^2^, *p* value = 0.25). However, daily vitamin B2 (riboflavin) and vitamin B6 intake was significantly higher among participants with higher caffeine intake (Table 1). Daily vitamin B2 intake was significantly higher in participants with more coffee consumption (Appendix A).

Participants with higher caffeine intake had a higher alpha diversity (Shannon index, *q* value < 0.001) (Figure 2). Alpha diversity did not differ by coffee intake (*q* = 0.19 for Shannon index). Bacterial beta diversity, i.e., bacterial community composition, also differed significantly based on the intake of caffeine (panel A) and coffee (panel B) (Figure 3).

We did not observe a significant difference in the relative abundance of bacterial phyla based on caffeine intake. At the family level, compared to lower caffeine intake, higher caffeine intake was related to a higher relative abundance of Ruminococcaceae (9.32% vs. 16.7%) and a lower relative abundance of Erysipelotrichaceae (4.25% vs. 1.56%) (*q* values < 0.05). At the genus level, higher caffeine intake was related to a higher relative abundance of *Faecalibacterium*, *Alistipes*, *Subdoligranulum*, and *Prevotella* but a lower relative abundance of *Erysipelatoclostridium* and *Lachnospiraceae (ASV0006)* (*q* values < 0.05) (Table 2). Higher coffee intake (more than 2 cups) was related to a higher relative abundance of *Faecalibacterium* and *Alistipes* (*q* values < 0.05) and a lower relative abundance of *Erysipelatoclostridium* (Table 2). Appendix A shows that many members of *Lachnospiraceae* family differed by caffeine intake.

In the multivariable negative binomial regression models, the incidence rate of having a non-zero count of *Alistipes* (IRR: 3.05, 95% CI: 1.10–8.48) and *Faecalibacterium* (IRR: 5.28; 95% CI: 2.68–10.4) was higher and the incidence rate of having a non-zero count of *Erysipelatoclostridium* (IRR: 0.07; 95% CI: 0.02–0.25) was lower in those who had higher caffeine intake compared to those who had lower caffeine intake. Higher coffee intake was associated with higher *Alistipes* (IRR: 2.84; 95% CI: 1.38–5.84) and *Faecalibacterium* (IRR: 2.35; 95% CI: 1.50–3.68) and lower *Erysipelatoclostridium* (IRR: 0.24; 95% CI: 0.07–0.84). Adjusting for total energy intake, BMI, and macronutrients in the analyses did not change the rate ratio estimate. However, the associations between *Faecalibacterium, Alistipes,* and caffeine and coffee intake were attenuated after adjustment of vitamin B2. There were also no significant associations between bacterial count and coffee intake. There was a statistically significant inverse association between *Erysipelatoclostridium* count and caffeine intake (IRR: 0.02; 95% CI: 0.003–0.17, *q* value < 0.05) (Table 3). In addition, adjusting for vitamin B6 in the model slightly attenuated the association between *Alistipes* count and caffeine intake, and adjusting for vitamin B12 in the model slightly attenuated the association between *Faecalibacterium* count and caffeine intake.

## 4. Discussion

This cross-sectional study showed that the colonic mucosa-associated bacteria differed significantly in the community composition and structure based on daily caffeine and coffee intake in adults. Higher caffeine and coffee intake was associated with higher richness and evenness of gut microbiota, a higher relative abundance of *Faecalibacterium* and *Alistipes*, and a lower relative abundance of *Erysipelatoclostridium* independent of multiple covariates. However, further multivariate analysis showed that vitamin B2 intake partially explained such observations. The association between *Erysipelatoclostridium* count and caffeine intake remained statistically significant after vitamin B2 intake was adjusted in the analysis. There was no significant association between bacterial count and coffee intake in multivariate models.

Our study showed that higher caffeine intake, but not coffee consumption, was associated with higher alpha diversity, i.e., richness and evenness of the gut bacteria in the colonic mucosa. In a Dutch population-based cohort study, coffee was shown to be associated with higher microbial alpha diversity. However, this study did not evaluate the association between phytochemicals in coffee and gut microbiota [26]. The beta diversity analysis showed a significant dissimilarity of bacterial community composition based on both caffeine intake and coffee consumption. Studies have associated lower alpha diversity with adverse health outcomes, such as a higher risk of colorectal cancer [27] and inflammatory bowel disease [28]. Whether caffeine could affect health outcomes through modulating gut microbiota biodiversity should be evaluated further.

We observed that higher caffeine intake and higher coffee consumption were related to a higher relative abundance of *Faecalibacterium.* A previous study showed increased *Faecalibacterium* species after mice fed a high-fat diet were treated with a caffeine-rich Chinese tea [29]. *Faecalibacterium* is a butyrate-producing bacterium [30] that has been shown to have positive impacts on human energy metabolism [31], and its anti-inflammatory properties have been shown [32]. Higher caffeine intake was also associated with a higher relative abundance of *Alistipes*, which belongs to Bacteroidetes phylum. *Alistipes* has been shown to be more abundant in populations with animal-based diets than in those with plant-based diets. It is more bile-tolerant and capable of protein breakdown [33]. *Alistipes* has been associated with both favorable and adverse health outcomes. Previous studies showed an inverse correlation between *Alistipes* and obesity [34], ulcerative colitis [35], and *Clostridium difficile* infection [36]. Conversely, an increased abundance of *Alistipes* has been associated with a higher frequency of abdominal pain in pediatric patients with irritable bowel syndrome [37]. It is noted that the adjustment of dietary vitamin B2 intake attenuated the association between caffeine and coffee intake and *Faecalibacterium* and *Alistipes*. Whether *Faecalibacterium* and *Alistipes* are independently modulated by caffeine in the colonic mucosa and the interaction between vitamin B2 and the colonic microenvironment should be further investigated.

Participants with lower caffeine intake had higher relative abundances of the Gram-positive obligate-anaerobe *Erysipelotrichaceae* family and *Erysipelatoclostridium* genus than those with higher caffeine intake. This observation remained statistically significant in multivariate analysis controlling vitamin B2 intake. *Erysipelatoclostridium* belongs to Firmicutes phylum. Higher *Erysipelatoclostridium* levels have been linked to diet-induced obesity in mouse models [38] and obesity in humans [39,40]. A previous study showed that patients with type 2 diabetes had higher *Erysipelotrichaceae* levels than patients with normal glucose tolerance [41]. There is growing evidence showing the adverse roles of *Erysipelotrichaceae* and *Erysipelatoclostridium* play in host lipid metabolism [42], immune response [43], inflammation [44], depression [45], metabolic-associated fatty liver disease [46], cancer [47], and response to cancer immune therapy [48,49]. However, in a human-based study, increased physical activity was associated with a higher relative abundance of *Erysipelotrichaceae* [50]. Nevertheless, given most studies have shown the detrimental effect of *Erysipelatoclostridium* on health, identifying pre- and probiotics that deplete this bacterium would be beneficial. It remains to be determined whether caffeine influences insulin resistance or metabolic diseases by modulating *Erysipelatoclostridium.*

As alluded to above, coffee has been shown to beneficially modulate human metabolic health by lowering the risks of obesity, type 2 diabetes, and metabolic syndrome [51,52,53]. The bioactive components in coffee include phenolic compounds (chlorogenic acids (CGAs)), alkaloids (caffeine and trigonelline), and diterpenes (cafestol and kahweol) [54]. One fecal microbial study of 147 patients found that polyphenol intake was similar between lower and higher coffee consumers. The author hypothesized that low coffee consumers could obtain polyphenols through other sources [10]. Several studies investigated the influence of caffeine, polyphenols, and CGAs on the gut microbiota. One in vitro colonic metabolism study found that coffee with the highest CGA levels induced a significant 0.5 logarithmic increase in the growth of *Bifidobacterium* spp. [55]. CGAs significantly increased the growth of *Bifidobacterium* spp. and *Clostridium coccoides/Eubacterium rectale* in a colonic microbiota culture model [55] and a human interventional study [56]. However, our study could not confirm the previous observation on *Bifidobacterium* and coffee intake [11]. The relative abundance of *Bifidobacterium* in the colonic mucosa was less than 0.50% in our study samples. Nevertheless, our study showed that the relative abundance of *Prevotella* was higher with a higher intake of caffeine and coffee. This observation was consistent with a human-based study that showed higher coffee consumers (45–500 mL/daily) had higher fecal *Prevotella* abundance [10]. However, it was not in line with an animal-based study that shows the opposite [9]. In summary, our preliminary study could not exclude the possibility that other nutrients in coffee, such as CGAs, polyphenols, and vitamin B family may have explained the association between coffee intake and gut microbiota composition and structure. Adjusting for dietary intake of vitamin B2 in the multivariate analysis did attenuate the association between gut bacteria and caffeine and coffee intake. It is noted that coffee has abundant vitamin B2.

Our study has uniqueness because we studied the mucosal-associated adherent microbiota using the colonic biopsies taken after bowel preparation. Compared to the luminal gut microbiota, the mucosa-associated adherent microbiota is more likely to interact directly with the immune cells in the mucosa and therefore impact host physiology [57]. Multivariate analysis was used to control for potential confounding effects of lifestyle and dietary factors. The study had some limitations. First, participants were mainly obese elderly or middle-aged white veterans, possibly affecting the generalizability of the findings to the non-veteran general population, women, lean individuals, or younger individuals. Second, study participants were retrospectively asked about their food consumption, which could introduce information bias. The caffeine intake was quantified by using 43 food items in the questionnaire. However, we could not exclude the possibility of measurement error in quantitating caffeine intake. Third, selection bias was likely as not all participants responded to the FFQ. Fourth, the relative abundance of multiple members in *Lachnospiraceae* family differed by caffeine intake. Metagenomic shotgun sequencing and metabolomics study should be conducted to define the bacterial species and their functions in association with coffee and caffeine intake. Lastly, the present analysis was based on a small sample size and was cross-sectional by nature. By virtue of these study limitations, the findings should be considered preliminary.

In summary, our study sheds light on the associations between caffeine intake and coffee consumption and the gut microbiota of individuals with endoscopically normal colons. The association between caffeine, gut microbiota, and health outcomes and the role of *Erysipelatoclostridium* in metabolic diseases deserves further investigation. In addition, the health effect of coffee may be partially explained by prebiotic vitamin B2. Future nutrigenomic and metabolomic studies are needed to identify who will benefit from coffee consumption. If we can understand how phytochemicals and nutrients maintain gut symbiosis and how they impact human physiology, we could uncover a novel mechanism for disease prevention.

## Figures and Tables

**Figure 1 nutrients-15-01747-f001:**
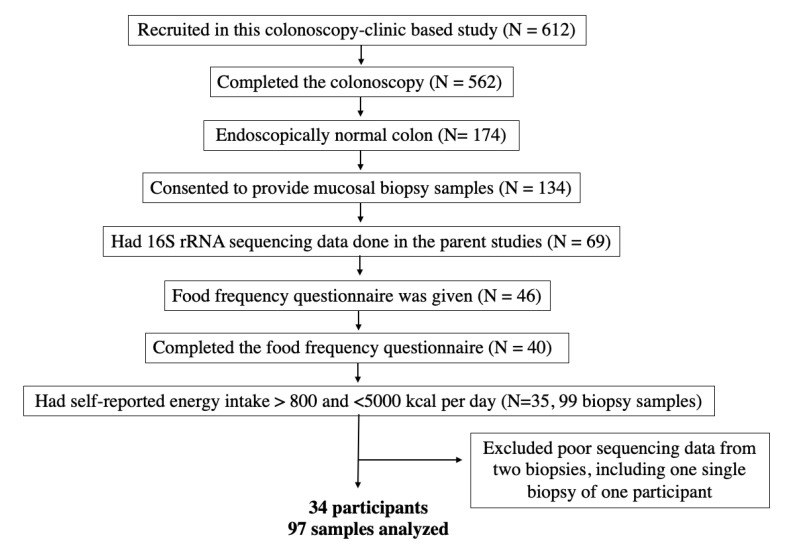
Flow chart of participant eligibility.

**Figure 2 nutrients-15-01747-f002:**
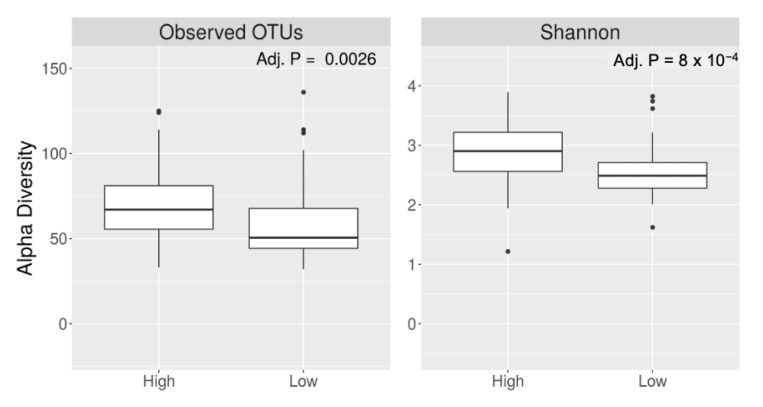
Alpha diversity of gut microbiota based on higher vs. lower caffeine intake.

**Figure 3 nutrients-15-01747-f003:**
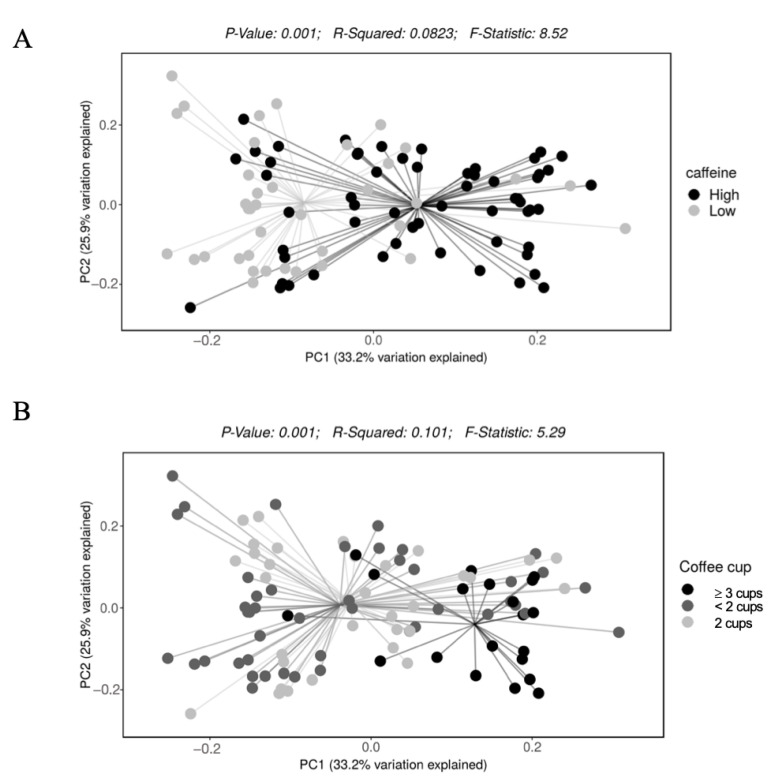
Beta diversity of gut microbiota based on daily intake of (**A**) caffeine and (**B**) coffee.

**Table 1 nutrients-15-01747-t001:** Basic characteristics of study participants based on caffeine intake.

Characteristics Mean ± Standard Deviation or *n* (%)	Low Caffeine < Median (*n* = 17)	High Caffeine ≥ Median (*n* = 17)	*p* Value
Caffeine (mg)	39.2 ± 6.4	138.9 ± 13.9	<0.0001
Age (years)	61.7 ± 1.3	62.2 ± 1.5	0.78
Men, *n* (%)	17 (100%)	16 (94%)	1.00
Racial Group			
Non-Hispanic white, *n* (%)	14 (82.4%)	14 (82.4%)	1.00
Body mass index (kg/m^2^)	32.6 ± 1.5	35.2 ± 1.6	0.25
Smoking status, *n* (%)			0.55
Never smokers	7 (41.2%)	6 (35.3%)	
Former smokers	8 (47.1%)	6 (35.3%)	
Current smokers	2 (11.8%)	5 (29.4%)	
Alcohol Status, *n* (%)			0.68
Never drinkers	4 (23.5%)	5 (29.4%)	
Former drinkers	4 (23.5%)	6 (35.3%)	
Current drinker	9 (53%)	6 (35.3%)	
Hypertension, yes, *n* (%)	12 (70.6%)	13 (76.5%)	1.00
Diabetes, yes, *n* (%)	8 (47.1%)	9 (52.9%)	1.00
Daily total calorie intake (kcal)	2039 (±557)	1748 (±813)	0.23
Total carbohydrate (grams/1000 kcal/day)	116 (±23.5)	111 (±17.5)	0.53
Total protein (grams/1000 kcal/day)	36.4 (±5.78)	40.0 (±8.37)	0.16
Total fat (grams/1000 kcal/day)	41.0 (±8.78)	43.4 (±4.96)	0.33
Vitamin B2 (mg/1000 kcal/day)	0.89 (±0.20)	1.28 (± 0.30)	0.0001
Vitamin B6 (mg/1000 kcal/day)	0.82 (±0.16)	1.02 (± 0.34)	0.02
Vitamin B12 (mcg/1000 kcal/day)	2.20 (±0.23)	2.76 (± 0.98)	0.06
HEI score ^1^	60.5 (±2.0)	61.4 (±2.4)	0.77

^1^ HEI, healthy eating index.

**Table 2 nutrients-15-01747-t002:** Relative abundance of bacterial genera by intake of caffeine and coffee.

	Lower Caffeine	Higher Caffeine		<2 Cups Coffee	2 Cups Coffee	≥3 Cups Coffee	
Bacterial Genus	Relative Abundance (%)		*q* Value	Relative Abundance (%)			*q* Value
*Erysipelatoclostridium*	3.14	0.10	<0.0001	1.42	1.79	0.19	0.22
*Faecalibacterium*	4.29	9.54	0.0003	5.66	5.26	15.16	<0.0001
*Lachnospiraceae (ASV0006)*	4.88	1.58	0.0007	3.46	2.56	2.03	0.33
*Alistipes*	0.57	1.32	0.01	0.46	1.31	1.84	<0.0001
*Subdoligranulum*	0.10	0.76	<0.0001	0.15	1.13	0.29	<0.0001
*Sutterella*	1.96	1.59	0.89	1.54	2.21	1.41	0.02
*Prevotella*	1.39	3.41	0.03	2.40	3.64	1.50	0.16

**Table 3 nutrients-15-01747-t003:** Multivariable negative binomial regression analysis of the association between caffeine and coffee intake and a non-zero bacterial count.

Genera	IRR (95% CI) ^1^	IRR (95% CI) ^2^	IRR (95% CI) ^3^
Caffeine			
*Alistipes*	3.17 (1.16–8.66)	3.05 (1.10–8.48)	1.40 (0.32–6.19)
*Faecalibacterium*	5.56 (2.84–10.9)	5.28 (2.68–10.4)	2.10 (0.85–5.20)
*Erysipelatoclostridium*	0.07 (0.02–0.25)	0.07 (0.02–0.25)	0.02 (0.003–0.17)
*Subdoligranulum*	0.85(0.49–1.48)	0.86 (0.52–1.42)	1.17 (0.57–2.42)
Coffee			
*Alistipes*	2.86 (1.37–5.97)	2.84 (1.38–5.84)	2.20 (1.00–4.89)
*Faecalibacterium*	2.39 (1.53–3.73)	2.35 (1.50–3.68)	1.49 (0.87–2.56)
*Erysipelatoclostridium*	0.26 (0.08–0.84)	0.24 (0.07–0.84)	0.31(0.07–1.36)
*Subdoligranulum*	1.08 (0.74–1.57)	0.97 (0.66–1.42)	1.10 (0.73–1.66)

CI: confidence interval. IRR: incidence rate ratio. ^1^ Negative binomial regression model for panel data used the rarefied bacterial count as the dependent variable. The reference group was lower intake of caffeine or coffee. Coffee intake was modeled as a continuous variable. The model was adjusted for age, race (non-Hispanic white, African American, and Hispanic), body mass index, smoking status (never, former, and current), alcohol use (never, former, and current), diabetes, hypertension, and colon segment. The incidence rate of having non-zero *Faecalibacterium* count was 4.56 times higher among those who had a higher intake of caffeine than those who had a lower intake. ^2^ Based on the first model, the model was further adjusted for HEI score. The incidence rate of having a non-zero *Faecalibacterium* count was 4.28 times higher among those who had a higher intake of caffeine than those who had a lower intake. ^3^ Based on the second model, the model was further adjusted for vitamin B2 intake. The incidence rate of having a non-zero *Faecalibacterium* count was 1.10 times higher among those who had a higher intake of caffeine than those who had a lower intake.

## Data Availability

Data available on request due to local policy on privacy.

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
