# Peer review of "The Association between Caffeine Intake and the Colonic Mucosa-Associated Gut Microbiota in Humans—A Preliminary Investigation"

_nutrients, 2023, doi:10.3390/nu15071747_

Round 1

Reviewer 1 Report

Manuscript ID: nutrients-2284706

Caffeine Intake and the Colonic Mucosa-Associated Gut Microbiota

The manuscript under appreciation describes the association between caffeine and coffee intake and the community composition and structure of human colonic adherent microbiota.

It is a work well done, the objectives are well defined and adequate methodology is performed. But it is a preliminary study of future works that could establish these described associations.

The manuscript can be accepted with minor revision.

The changes that are suggested to improve the work are the following:

-          The title should be changed to mention the association between caffeine consumption and the microbiota. This would clarify what is going to be read in the text.

-          In lines 253-254: “Whether the beneficial effect on caffeine is mediated by caffeine should be further investigated”. This sentence is not understood, it must be reviewed.

-          In lines 297-298 “The adherent microbiota may be more related to host physiology than fecal microbiota”. This sentence this sentence should be explained.

-          It is interesting to make the consideration that the authors make about the possible involvement of other bioactive components (polyphenols, CGA and vitamins) in this modulation of the microbiota. Therefore it should be considered in the abstract. In this way, the study would not focus only on estimating the amount of caffeine ingested as the only factor to be considered.

The preliminary or previous character of the study should appear in the abstract. Since exhaustive studies are necessary to corroborate the involvement of caffeine in the relative abundance of certain genera of bacteria, although , the authors consider it at the end of the study.

Author Response

Manuscript ID: nutrients-2284706

Title in the revision: The Association between Caffeine Intake and the Colonic Mucosa-Associated Gut Microbiota in Humans – a Preliminary Investigation

We appreciate reviewers’ insightful comments on our manuscript. Please see the point-by-point response below. In addition, we have proofread the revised manuscript. Thank you!

Li Jiao, MD, PhD

Reviewer 1:

-          The title should be changed to mention the association between caffeine consumption and the microbiota. This would clarify what is going to be read in the text.

Response: The title has been updated to “The Association between Caffeine Intake and the Colonic Mucosa-Associated Gut Microbiota in Humans – a Pilot Investigation”.

-          In lines 253-254: “Whether the beneficial effect on caffeine is mediated by caffeine should be further investigated”. This sentence is not understood, it must be reviewed.

Response: Thank you for the comment. It should be stated as “Whether the beneficial effect of coffee on health outcomes is mediated by caffeine should be further investigated.”

-          In lines 297-298 “The adherent microbiota may be more related to host physiology than fecal microbiota”. This sentence should be explained.

Response: This sentence has been changed to “Compared to the luminal gut microbiota, the mucosa-associated adherent microbiota is more likely to interact directly with immune cells and therefore host physiology. One citation is added to the revision, reference 47.

-          It is interesting to make the consideration that the authors make about the possible involvement of other bioactive components (polyphenols, CGA and vitamins) in this modulation of the microbiota. Therefore it should be considered in the abstract. In this way, the study would not focus only on estimating the amount of caffeine ingested as the only factor to be considered.

Response: We added one sentence to the end of the Abstract: “Our preliminary study could not evaluate other phytochemicals or prebiotics in coffee”. Thank you.

The preliminary or previous character of the study should appear in the abstract. Since exhaustive studies are necessary to corroborate the involvement of caffeine in the relative abundance of certain genera of bacteria, although , the authors consider it at the end of the study.

Response: The abstract has been updated.

Reviewer 2:

Overall Impression:  The manuscript presents results of an interesting study of caffeine intake and changes in the colonic mucosa-associated gut microbiota. The authors conclude that higher caffeine and coffee intake is associated with higher alpha diversity. Higher relative abundance and count of Faeclibacterium and Alsitipes while there was lower relative abundance and count of Erysipelatoclostridium.  This was related to high intake of caffeine and coffee.

Response: Thank you.

Critique:  In their Discussion and Conclusions the authors point out that their study has a number of limitations, all of which I agree with.

  1. Participants were mainly obese elderly or middle-aged white veterans
  2. Study participants were retrospectively asked about their food consumption which depends on accurate recall.
  3. Selection bias was likely as not all participants responded to the Food Frequency Questionaire (this is a particular weakness that I object to)
  4. The relative abundance of multiple members in Lachnospiraceaefamily differed by caffeine intake.  Metagenomic shotgun sequencing and meabolomics study should be conducted to define the bacterial species and their functions in association with coffee and caffeine intake
  5. The present study was based on a small sample size and was cross-sectional by nature.

Thus the Authors concluded that their findings should be considered preliminary.

With the last caveat, I believe the study they presented was as controlled as  much as possible.  Therefore if the title can be changed to reflect that this is a preliminary study, I would be in favor of considering publication.

Response: Thank you for the comment. The title has been changed to “The Association between Caffeine Intake and the Colonic Mucosa-Associated Gut Microbiota in Humans – a Preliminary Investigation”.

Reviewer 2 Report

Critique Nutrients Manuscript 2284706

Caffeine Intake and the Colonic Mucosa-Associated Gut Microbiota

Annie Dai, Kristi Hoffman, Anthony A. Xu, et al.

Overall Impression:  The manuscript presents results of an interesting study of caffeine intake and changes in the colonic mucosa-associated gut microbiota. The authors conflude that higher caffeine and coffee intake is associated with higher alpha diversity. Higher relative abundance and count of Faeclibacterium and Alsitipes while there was lower relative abundance and count of Erysipelatoclostridium.  This was related to high intake of caffeine and coffee.

Critique:  In their Discussion and Conclusions the authors point out that their study has a number of limitations, all of which I agree with.

1.    Participants were mainly obese elderly or middle-aged white veterans

2.   Study participants were retrospectively asked about their food consumption which depends on accurate recall.

3.   Selection bias was likely as not all participants responded to the Food Frequency Questionaire (this is a particular weakness that I object to)

4.   The relative abundance of multiple members in Lachnospiraceae family differed by caffeine intake.  Metagenomic shotgun sequencing and meabolomics study should be conducted to define the bacterial species and their functions in association with coffee and caffeine intake

5.   The present study was based on a small sample size and was cross-sectional by nature.

Thus the Authors concluded that their fundings should be considered preliminary.

With the last caveat, I believe the study they presented was as controlled as  much as possible.  Therefore if the title can be changed to reflect that this is a preliminary study, I would be in favor of considering publication.

Author Response

(The authors gave the same response as above.)

Round 2

Reviewer 2 Report

The authors have now labeled this a preliminary study, which indeed it is.  I am comfortable at this point with suggesting that it be published.

Author Response

Addressing further comments. 2284706

March 26, 2023

Dear Editor:

Thank you for the additional comments. We believe that these further comments improved the soundness of our research article.

 1) Differences in gut microbiota between the two groups may be due to differences in BMI between the two groups.

Response: Thank you for the comment. Table 1 shows that there is no significant difference in BMI by caffeine intake (P value is 0.25). In addition, the adjustment of BMI in the multivariate models did not change the rate ratio.  The first paragraph of Result has been updated in the revised manuscript.

2) Were there any differences in nutrients other than coffee intake between the two groups? For example, total calories, protein, fat, and carbohydrate intakes.

Response: Thank you for the comment. In our previous version, HEI was adjusted in the model for controlling the confounding effect of dietary quality. According to this comment, we further evaluated other macro- and micro-nutrients as confounding factors. As shown in our previous study, dietary intake of vitamin B2 was associated with Faecalibacterium, Alistipes, and Erysipelatoclostridium (PMC6470629). In fact, the intake of vitamins B2 and B6 differed by caffeine intake (revised Table 2). Therefore, vitamin B2, B6, and B12 was adjusted in the mode individually. As the updated result showed, adjustment of vitamin B2 attenuated the association between caffeine and coffee intake and Faecalibacterium and Alistipes count. Adjusting for vitamin B6 in the model slightly attenuated the association between Alistipes count and caffeine intake and adjusting for vitamin B12 in the model slightly attenuated the association between Faecalibacterium count and caffeine intake. The association between Erysipelatoclostridium count and caffeine remained statistical significant. Please see last paragraph of Result. 

All the related updates have been made in Abstract, Introduction, Method, Results, Discussion, and Tables. Please see the track-change version of the revision for all the changes. We also included more discussion on Erysipelatoclostridium to respond to the request to increase the word. The entire manuscript is now more than 4000 words including abstract.

Thank you.

Li Jiao, MD. PhD
